# Reduced Corneal Innervation in the CD25 Null Model of Sjögren Syndrome [note 1]

**DOI:** 10.3390/ijms19123821

**Published:** 2018-11-30

**Authors:** Mary Ann Stepp, Sonali Pal-Ghosh, Gauri Tadvalkar, Alexa R. Williams, Stephen C. Pflugfelder, Cintia S. de Paiva

**Affiliations:** 1Department of Anatomy and Regenerative Biology, The George Washington University School of Medicine and Health Sciences, Washington, DC 20037, USA; spghosh@gwu.edu (S.P.-G.); grtdv@email.gwu.edu (G.T.); arwilliams@email.gwu.edu (A.R.W.); 2Department of Ophthalmology, The George Washington University School of Medicine and Health Sciences, Washington, DC 20037, USA; 3Department of Ophthalmology, Ocular Surface Center, Cullen Eye Institute, Baylor College of Medicine, Houston, TX 77030, USA; stevenp@bcm.edu (S.C.P.); cintiadp@bcm.edu (C.S.d.P.)

**Keywords:** Sjögren syndrome, corneal sensitivity, autophagy, CD25 KO

## Abstract

Decreased corneal innervation is frequent in patients with Sjögren Syndrome (SS). To investigate the density and morphology of the intraepithelial corneal nerves (ICNs), corneal sensitivity, epithelial cell proliferation, and changes in mRNA expression of genes that are involved in autophagy and axon targeting and extension were assessed using the IL-2 receptor alpha chain (CD25 null) model of SS. ICN density and thickness in male and female wt and CD25 null corneas were assessed at 4, 6, 8, and 10/11 wk of age. Cell proliferation was assessed using ki67. Mechanical corneal sensitivity was measured. Quantitative PCR was performed to quantify expression of beclin 1, LC3, Lamp-1, Lamp-2, CXCL-1, BDNF, NTN1, DCC, Unc5b1, Efna4, Efna5, Rgma, and p21 in corneal epithelial mRNA. A significant reduction in corneal axon density and mechanical sensitivity were observed, which negatively correlate with epithelial cell proliferation. CD25 null mice have increased expression of genes regulating autophagy (beclin-1, LC3, LAMP-1, LAMP-2, CXCL1, and BDNF) and no change was observed in genes that were related to axonal targeting and extension. Decreased anatomic corneal innervation in the CD25 null SS model is accompanied by reduced corneal sensitivity, increased corneal epithelial cell proliferation, and increased expression of genes regulating phagocytosis and autophagy.

## 1. Introduction

CD25 is the interleukin-2 receptor α chain (IL2RA). Together with the β (IL2RB) and the γ (IL2RG) chains, it constitutes the high-affinity IL2 receptor (IL2R), which is expressed at high levels on T regulatory (Treg) cells [1,2]. The binding of the cytokine IL-2 to the IL-2 receptor triggers signal transduction via the JAK pathway by upregulating STAT signaling and inflammation. While rare, the literature describes several patients deficient for CD25 expression on their Tregs, with varying degrees of immune dysregulation [2,3,4,5,6]; a recent study highlights the ocular symptoms (itching, burning, foreign body sensation, photophobia) of a patient whose Tregs are deficient in the expression of CD25 [7]. Tregs suppress inflammation and regulate autoimmunity, both systemically [1] and at the ocular surface [8]. Enhanced autoimmunity is associated with ocular surface pathology and dry eye disease (DED) [9,10,11,12]. Previous studies have shown that CD25 null mice develop a severe form of dry eye disease similar to Sjögren syndrome [13,14,15]; CD25 null lacrimal glands are severely inflamed and they show increased numbers of apoptotic cells [16]. The corneas of these mice show reduced smoothness, increased permeability, increased infiltration of CD4+ T cells, and increased expression of IL-6, IL-17, and IL-21 mRNAs [15,16].

The impact of the ocular surface pathology seen in CD25 null mice on the density and function of the intraepithelial corneal nerves (ICNs) that innervate the corneal surface is not known. These nerves trigger tear production and the blink response and they are critical in maintaining a stable tear film over the ocular surface [17,18]. Complete denervation of the ICNs leads to the cessation of corneal epithelial cell proliferation and the differentiation of limbal stem cells [19,20]. In addition, turnover of axonal fragments derived from ICNs generates debris that is phagocytosed by corneal epithelial cells [17]. The transient receptor potential (TRP) family of cation channels are present on sensory axons and they are activated by molecules present in tears [21,22]; the leaky epithelial barrier in CD25 null corneas [16] could activate the ICNs.

Dry eye disease is associated with changes in corneal sensory nerve morphology and dysfunction in humans [23,24] and mice [25,26,27]. In this study, we show how the ICNs and the corneal epithelium cope with the loss of CD25. We quantify ICN density, thickness, corneal sensitivity, corneal epithelial cell proliferation, and changes in expression of axon targeting and autophagy RNAs in CD25 null corneas.

## 2. Results

### 2.1. CD25 Null Mice Have Reduced Axon Density and Increased Corneal Epithelial Cell Proliferation as Early as Four Weeks after Birth

To determine whether the loss of CD25 impacts intraepithelial corneal nerves (ICNs) in mice, we assessed axon density in male and female CD25 null mice and wt littermates as a function of time after birth by localizing neuron specific βIII tubulin in whole flat mounted corneas and quantifying axon density using Sholl analysis. Representative images from male and female wt and CD25 null at 4, 6, 8, and 10/11 wk are shown in Figure 1. At 4 wk, in both wt and CD25 null corneas, clusters of ICNs are seen emerging from the stroma and extending towards the corneal center; a vortex has not yet formed. By 6 wk the vortex is present in both wt and CD25 null corneas.

Quantification of axon density data focusing on sex and genotype are presented in Figure 2. No significant differences emerge between sexes when wt and CD25 null data for male and female corneas are analyzed by non-parametric ANOVA (Kruskal-Wallis). While a trend for females to have higher axon density is seen, combining male and female axon density data for all ages to increase the power of the analysis, yields *p* values that are greater than 0.05 for both genotypes of mice.

In Figure 3A, data from both male and female mice are combined and wt and CD25 null mice are compared to one another using non-parametric Mann-Whitney *t* tests. Data for each genotype are compared over time by non-parametric ANOVA. Axon density increases significantly as assessed by ANOVA as wt corneas mature with the increase becoming significant for 8 and 10/11 wk as compared to 4 wk (black bar and asterisk). Axon density in CD25 null corneas increased between 4 and 6 wk (grey bar and asterisk), but thereafter remained the same. CD25 null mice have significantly lower axon density at 4, 8, and 10/11 wk of age when compared to wt mice (maroon bar and asterisk). While CD25 null mice also show lower axon density at 6 wk, the difference is not significant. Also shown in Figure 3A is a comparison between genotypes after combining data for all four ages assessed. The difference between wt and CD25 null axon density is significant and it shows a reduction of approximately 50% in null corneas.

Surgical denervation of the ICNs leads to a dramatic reduction in corneal epithelial cell proliferation [20], whereas chronically elevated corneal epithelial cell proliferation destabilizes the ICNs and leads to reduced axon density [28]. To determine whether the reduced axon density seen in CD25 null corneas is impacted by differences in corneal epithelial cell proliferation, we next assessed the numbers of ki67+ corneal epithelial cells in the same corneas used for the determination of axon density. While epithelial cell proliferation in wt corneas increases between 4 and 6 wk, the increase is not significant by ANOVA; however, between 6 and 8 wk, wt corneal epithelial cell proliferation decreases (black bar and asterisk). Cell proliferation in CD25 null corneas does not vary as the cornea matures. Using t tests to directly compare genotypes, CD25 null mice have significantly higher corneal epithelial cell proliferation at 4 and 10 wk of age as compared to wt mice (maroon bars and asterisks). Also shown in Figure 3B is a comparison between genotypes after combining ki67 data for all four ages assessed. The increase in cell proliferation in CD25 null corneas is significant but minimal when compared to the reduction that was seen in axon density.

Next, we calculated the Spearman correlation coefficient (r) for paired data for axon density and cell proliferation from wt and CD25 null corneas at 4, 6, 8, and 10/11 wk, as well as for all ages combined, as shown in Figure 3C. For wt corneas, the Spearman *r* value is negative at 4, 6, and 10/11 wk and was statistically significant at 6 and 10/11 wk of age. The Spearman *r* value is positive only at 8 wk; this time point corresponds to the reduction cell proliferation rate seen between 6 and 8 wk as corneas reach maturity. Combining all ages that were assessed results in a negative, significant correlation between axon density and cell proliferation in wt corneas. These data confirm in the healthy wt mouse cornea, higher rates of cell proliferation correlate with lower axon densities. In CD25 null corneas, the Spearman *r* value is negative at all ages assessed and is statistically significant at 4 and 8 wk of age. Combining all ages also results in a negative significant correlation between axon density and cell proliferation in CD25 null corneas. While the surgical denervation of all sensory axons reduces corneal epithelial cell proliferation [19,20], the partial denervation that was seen in CD25 null corneas correlates with elevated corneal epithelial cell proliferation.

### 2.2. CD25 Null Corneas Have Fewer Stromal Axons

Stromal nerves enter the cornea from the trigeminal ganglion at the limbus; ICNs branch from these stromal nerves and enter the epithelium. Debridement wounding, where the epithelium is removed and the basement membrane is left intact, leads to a reduction in the nerves within the stroma [29]. To determine whether the reduction in corneal epithelial sensory nerves that was seen in CD25 null corneas is secondary to reduced numbers of stromal nerves, we next quantified the stromal nerve arborization in 4 and 8 wk wt and CD25 null corneas. Representative images and their quantification using NeuronJ are shown in Figure 4. Data show that, while stromal nerves are reduced as compared to wt in CD25 null corneas at 4wk, the reduction is not significant; by 8 wk, stromal nerves are significantly less abundant in the stroma of CD25 null corneas. These data show that stromal nerve arborization in CD25 null corneas is reduced to approximately 70% that seen in wt mice of the same age and sex. ICN density is 50% of that seen in wt corneas.

### 2.3. CD25 Null Corneas Have Reduced Corneal Sensitivity but Increased Apical Extension and Branching of Their Intraepithelial Nerve Terminals

Since CD25 null mice have fewer stromal nerves and reduced ICN axon density, we next assessed the sensitivity of their corneas to mechanical touch, as shown in Figure 5A. Whereas, 6–8 wk female C57BL/6J and the CD25 wt littermates had mechanical corneal thresholds of 2 cm, 6–8 wk female CD25 null mice had values of 1.4 cm, a difference that is significant and indicates reduced sensation. ICNs run parallel to the basement membrane and extend axon branches perpendicularly between corneal epithelial basal cells from the basal aspect of the basal cells towards the apical most squames; these axons are referred to as intraepithelial nerve terminals (INTs). As INTs approach the tight junctions in the apical squames, they turn and continue extending parallel to the ocular surface. Using three-dimensional (3D) confocal imaging, we next assess INT apical extension by quantifying pixel intensity in cross sections generated from confocal image stacks showing βIII tubulin positive ICNs. Representative en face confocal image projection stacks and 3D generated cross sections are shown in Figure 5B. In Figure 5C, we show that sub-basal axon thickness is significantly less in CD25 null corneas.

CD25 null corneas have fewer, thinner ICNs than wt corneas. In Figure 5D, a schematic image showing how the INT data were generated is presented. In addition, data showing that while there are significantly fewer INTs in the basal cell layer, there are more INTs in the apical layers in CD25 null corneas. Thus, while there are fewer ICNs, once INTs form, they extend apically and elongate more than those from wt corneas.

### 2.4. Loss of Axon Density and Reduced Corneal Sensitivity in CD25 Null Mice is Accompanied by Increased Corneal Epithelial Expression of mRNAs Encoding Proteins That Mediate Autophagy and Phagocytosis

The reduced axon density and corneal sensitivity in CD25 null corneas could be secondary to a number of factors. To study the mechanisms underlying the loss of axons in CD25 null corneas, corneal epithelium of 6–8 wk wt and CD25 null female mice was harvested by debridement and was used to isolate RNA. qPCR studies were performed using primers for mRNAs whose expression was associated with increased phagocytosis and autophagy as well as axon targeting and extension. Data are presented in Figure 6 after normalization against mRNA expression in wt mice. We find no significant differences in expression in CD25 null corneal epithelium for several mRNAs whose proteins positively mediate axon targeting, including netrin1 and its receptors DCC, Unc5b, and Efna5. In addition, the expression of Repulsive guidance molecule a (Rgma), a protein that negatively regulates axon targeting, was not altered in CD25 null corneas. However, we do see the up-regulation of several proteins whose expression is altered in cells undergoing increased phagocytosis and autophagy, including Beclin1, LC3, LAMP1, LAMP2, and Brain derived neurotrophic factor (BDNF). While CXCL1 is also elevated, the increase was not significant. Taken together, these data indicate that loss of CD25 leads to reduced numbers of intraepithelial corneal nerves and reduced corneal sensitivity, in part, due to increased proliferation and phagocytosis of ICNs by corneal epithelial cells. Reduced axon density in CD25 null mice does not appear to be due to reduced recruitment and targeting of axons to the apical-most cell layers by the corneal epithelium.

## 3. Discussion

The Interleukin-2 receptor alpha chain (CD25) is expressed primarily on the surfaces of activated T and B cells and it is one of the biomarkers of T regulatory cells (Tregs) [1,2]. Loss of expression of CD25 is associated with the dysregulation of Tregs, autoimmunity, and ocular surface pathology [3,4,5,6]. While CD25 has been reported to be expressed on conjunctiva and cornea of the C57BL/6 mouse strain [30,31] and on human ocular surface epithelia [31], RNAseq studies show that CD25/IL2RA mRNA is undetectable in the Balb/c unwounded corneal epithelium and stroma [32]. While the ability of CD25-expressing Tregs to regulate autoimmunity make it likely that the pathology that develops in CD25 null mice is due to altered immune cell homeostasis, since the mouse lacks CD25 on all of its cells and tissues, we can not rule out the possibility that reduced localization of CD25 on ocular surface epithelia or within corneal stromal cells could also play a role. Inflammation in the CD25 null cornea increases over time [13,15]; at 4 wk, CD4+ and CD8+ cells have yet to infiltrate the CD25 null stroma at numbers greater than controls [15]. Since stromal nerves and ICNs are both altered significantly at 4 wk in CD25 null corneas, cellular immune factors are unlikely to contribute to the ocular surface pathology that was seen in CD25 null mice.

Previous studies on CD25 null mice show increased corneal permeability and increased corneal epithelial basal cell apoptosis by 8–12 wk of age [15]. A sustained chronic inflammatory environment in the corneal stroma could disrupt the targeting of sensory nerves from the trigeminal ganglion. Proteases that were released by activated immune cells release cytokines and growth factors that bind to heparan sulfate proteoglycans (HSPGs) [33]. Netrin-dependent guidance of axons has been shown, in C. elegans, to be regulated by HSPGs, including syndecan-1 and glypican [34,35]. Proteases also shed ectodomains of integrins [29] and growth factor receptors [36] destabilizing axons and reducing their bundling leading to thinner more fragile axons. As early as 4 wk after birth, stromal nerves are also reduced in CD25 null corneas; the reduction in stromal nerves persists through 8 wk of age. Fewer stromal nerves are available to innervate the corneal epithelium. The CD25 null corneal epithelial cells produce similar levels of mRNAs for proteins that attract axons and axons branch efficiently to the apical aspect of the epithelium. Yet, the combination of fewer and thinner ICNs yields corneas with reduced cornea sensitivity.

The impact the loss of CD25 has on corneal epithelial cell homeostasis is complex. Previous studies showed that CD25 null corneal epithelial basal cells show increased rates of apoptosis [15], and, shown here, corneal epithelial cell proliferation increases. Under homeostasis, ICNs are supported by corneal epithelial cell membranes that wrap around them and function like glial cells to protect them from forces exerted as neighboring corneal epithelial cells undergo cell division, apoptosis, and terminal differentiation [17]. Increased cell turnover contributes to reduced ICN density by increasing the mechanical load that was applied to axon membranes by corneal epithelial cells.

CD25 null corneal epithelial cells increase their expression of mRNAs for autophagy proteins. The corneal epithelial cells remove debris that is generated by the normal shedding of ICN axon tips by phagocytosis [17]. Recent studies have shown the involvement of autophagy proteins in a non-canonical pathway, called LC3-associated phagocytosis (LAP). In LAP, autophagy machinery is recruited to phagosomes to permit the removal of dead cells and debris without inducing an immune response [37,38,39]. Preventing LAP by removing one or more autophagy associated proteins induces a lupus-like autoimmune condition in mice [37]. LAP also plays an important role in the removal of shed optic discs by RPE cells after their ligation by integrins [40]. Byun and colleagues have shown increased expression of autophagy markers in tears and impression cytology extracts from patients with dry eye disease, as well as from 16 wk NOD/LtJ mice [41] The increased expression of the autophagy proteins LC3, Beclin1, LAMP1, LAMP2, and BDNF in CD25 null corneal epithelium indicate that the corneal epithelium is either phagocytosing or attempting to phagocytose excess axonal and apoptotic cell debris. In aged C57BL6 mouse with dry eye disease and reduced axonal density, expression by corneal epithelial cells of the same autophagy associated mRNAs is not altered [27].

## 4. Methods

### 4.1. Animals

The research protocol was approved by the Baylor College of Medicine Institutional Animal Care and Use Committee, and it conformed to the standards in the ARVO Statement for the Use of Animals in Ophthalmic and Vision Research. Heterozygous breeding pairs of the CD25 null (B6;129S4-*IL-2ra*^tm1Dw^/J) mice were purchased from The Jackson Laboratory (Bar Harbor, ME, USA) for the establishment of breeding colonies and raised under specific pathogen free conditions in the standard vivarium.

CD25 null mice of both sexes were used at four, six, eight, and ten to eleven wk. Wild-type littermates from the colony were used as controls. Both eyes from each mouse were used; therefore, the number of corneas assessed is twice the number of mice used. For CD25 null mice, we assessed axon density on the following numbers of corneas: 12 male and 11 female corneas at four weeks (wk), nine male and 18 female corneas at 6 wk, 17 male and 11 female corneas at 8 wk, and 12 male and 14 female corneas at 10/11 wk. For strain matched wt control mice, we assessed the following numbers of corneas: six male and six female corneas at 4 wk, 14 male and eight female corneas at 6 wk, 11 male and nine female corneas at 8 wk, and 10 male and eight female corneas at 10/11 wk. The number of corneas used varies due to differences in the number of males and female mice arising during breeding and a desire to use all of the mice. Female C57BL/6J mice aged 6–8 wk of age were purchased from Jackson Laboratories (Bar Harbor) and were used in the corneal mechanical sensitivity test.

### 4.2. Corneal Mechanical Sensitivity

Corneal sensitivity was measured in 6–8 wk old wt and CD25 null female mice under a surgical loupe with a 9-0 nylon monofilament of different lengths (1.0, 1.5, 2.0, 2.5, 3.0, 4.0 cm) as previously published [27]. While holding the animal, a nylon filament was applied to the cornea and a positive response was indicated by a clear stimulus-evoked blink and retraction of the eye into the ocular orbit. The central cornea was tested six times at each filament length. The response was considered positive when the animal blinked more than or equal to 50% the number of times tested. If a blink response could not be elicited at a monofilament length of 1.0 cm, corneal sensitivity was recorded as 0.

### 4.3. Immunofluorescence

Fixing and staining of mouse corneas for the identification of the intraepithelial nerve terminals has been described previously [28]. Corneas were incubated with βIII tubulin (Biolegend, #801201) and/or ki67 (Abcam, # ab16667), at 4 °C overnight.

### 4.4. Microscopy

Confocal microscopy was performed at the GW Nanofabrication and Imaging Center at the George Washington University Medical Center. For axon density by Sholl (25 × magnification) and axon thickness (40 × magnification) analysis, images were acquired using the Zeiss Cell Observer Z1 spinning disk confocal microscope (Carl Zeiss, Inc., Thornwood, NY, USA), equipped with ASI MS-2000 (Applied Scientific Instrumentation, Eugene, OR, USA) scanning stage with z-galvo motor, and Yokogawa CSU-X1 spinning disk. A multi-immersion 25 ×/0.8 and 40 ×/1.4 objective lens, LCI Plan-Neofluor, was used for imaging, with oil immersion. Evolve Delta (Photometrics, Tucson, AZ, USA) 512 × 512 EMCCD camera was used as detector (80-msec exposure time). A diode laser emitting at 568 nm was used for excitation (54% power). Zen Blue software (Carl Zeiss, Inc., Oberkochen, Germany) was used to acquire the images, fuse the adjacent tiles, and produce maximum intensity projections. The adjacent image tiles were captured with overlap to ensure proper tiling. All images were acquired while using the same intensity settings. Sholl analysis was performed using ImageJ, as described previously [42]. For high-resolution immunofluorescence imaging, a confocal laser scanning microscope (Zeiss 710) that was equipped with a krypton-argon laser was used to image the localization of Alexa Fluor 594 (568 nm excitation; 605/32 emission filter). Optical sections (*z* = 0.5 μm or 1 μm) were acquired sequentially with a 63x objective lens. For the quantification of intraepithelial nerve terminals, 3D images were rotated to generate cross section views using Volocity software (Version 6.3, Perkin Elmer, Waltham, MA, USA) and images were presented as cross sections projected through the length of the acquired image (135 μm). Pixel intensity data for apical and basal projecting nerve fibers were obtained, as described previously [43], for no fewer than four corneas per variable. For studies showing stromal nerve arborization, NeuronJ was used. NeuronJ is publically available (https://imagescience.org/meijering/software/neuronj/) and it was used as described [44]. We used female mice at 4 and 8 wk of age; stromal nerve arborization data were less variable between individual corneas than sensory nerve density data, which allowed us to use fewer corneas for these assessments.

### 4.5. qPCR

For quantitative polymerase chain reaction (qPCR) studies, epithelium was scraped using a dulled blade and was frozen immediately in liquid N2. Four corneas per sample and at least three samples per time point were used for the qPCR studies. RNA was extracted using a QIAGEN RNeasy Plus Micro RNA isolation kit (Qiagen, Germantown, MD., USA), following the manufacturer’s protocol. Two mice per age were pooled and RNA was isolated. After isolation, the concentration of RNA was quantified using a NanoDrop^®^ ND-2000 Spectrophotometer (Thermo Scientific, Wilmington, DE, USA) and stored at −80 °C until used. qPCR was performed using a Bio-Rad CFX384 Real-Time PCR detection system. The primers used were ordered from Bio Rad, unless otherwise specified: Bec1(qMmuCID0005981), LC3 (qMmuCED0045817), LAMP1 (qMmuCID0027030), LAMP2 (qMmuCID0011408), CXCL1 (qMmuCED0047655), BDNF (qMmuCED0050333), NTN1 (Qiagen #QT00128478), DCC (Qiagen #QT00135100), Unc5b (Qiagen #QT00167846), Efna4 (Qiagen #QT00100681), Efna5 (Qiagen #QT00116494), Rgma (Qiagen #QT00310583), and GAPDH (qMmuCED0027497). qPCR data are normalized against GAPDH.

### 4.6. Statistical Analyses

Quantitative data are presented as mean ± standard error of the mean. All statistical tests were performed using the GraphPad Prism Program, Version 6 (GraphPad Software, Inc. San Diego, CA, USA). A *p* value < 0.05 was considered to be statistically significant. Data are analyzed using one-way ANOVA to allow for the comparison of data from different age mice (4, 6, 8, 10/11) or using *t* tests to compare genotypes (wt and CD25 null). When means are distributed normally, parametric testing is performed; otherwise, non-parametric testing is performed.

## 5. Conclusions

The loss of CD25 increases the expression of genes regulating non-canonical LC3-associated phagocytosis by corneal epithelial cells. While this may permit the corneal epithelum to more efficiently remove apoptotic corneal epithelial cells and axonal debris, it also may contribute to increased epithelial cell apoptosis, proliferation, altered sensory nerve morphology, and reduced sensory nerve function. Further studies are needed to sort out the mechanisms underlying the pathology that was seen in this mouse model for Sjögren syndrome.

## Figures and Tables

**Figure 1 ijms-19-03821-f001:**
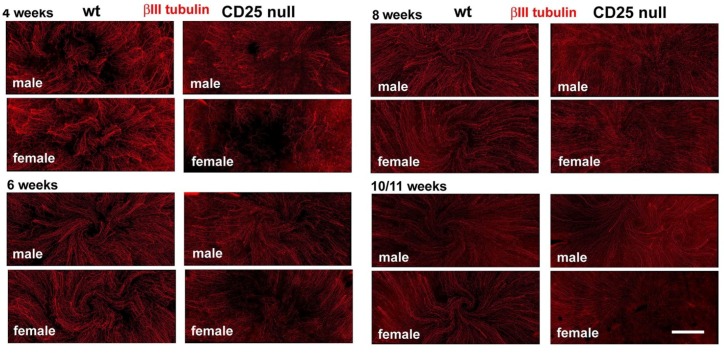
Intraepithelial corneal nerves (ICNs) in male and female wt and CD25 null corneas during early corneal maturation. Shown are representative confocal images from wt land CD25 null male and female corneas obtained, as described in the methods section at 4, 6, 8, and 10/11 wk after staining with antibodies against βIII tubulin. Bar = 500 µm.

**Figure 2 ijms-19-03821-f002:**
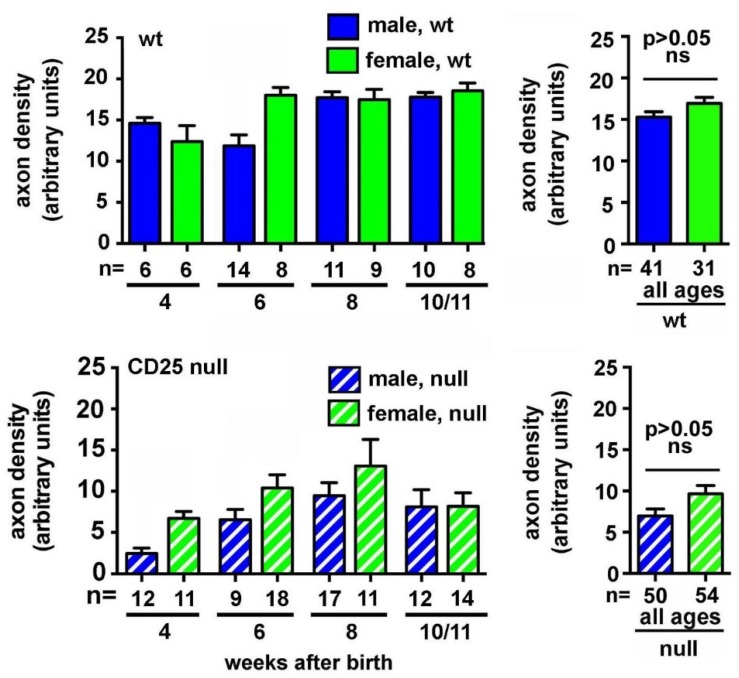
ICN density does not vary significantly between male and female wt and CD25 null mice. ICN density was quantified by Sholl analysis, as described in the methods section. No significant differences are seen between male and female mice at any of the ages studied. Despite a trend for female corneas to have higher axon densities, performing non-parametric t tests on data for males and females after combining all ages also yields *p* values above 0.05 for both genotypes.

**Figure 3 ijms-19-03821-f003:**
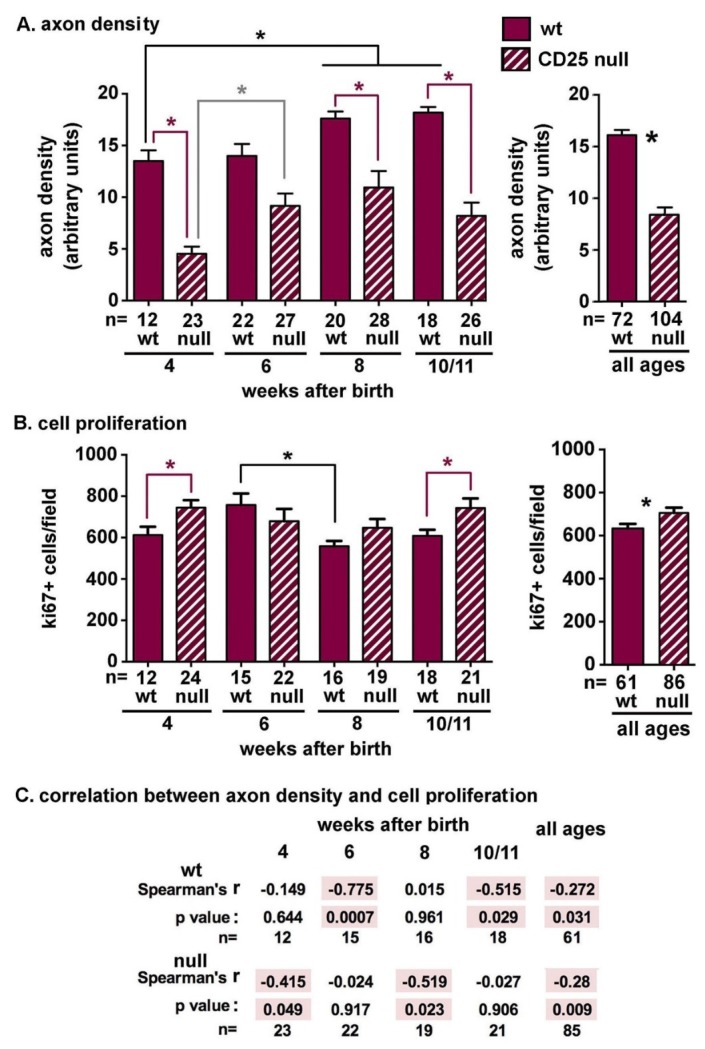
ICN density is reduced and corneal epithelial cell proliferation is increased in CD25 null corneas. (**A**) ICN density for male and female corneas were combined to allow us to determine the differences between genotypes and over time as corneas mature. The numbers of corneas used for each assessment are shown beneath each bar. Using ANOVA to compare the axon densities for all ages (4, 6, 8, 10/11) for wt corneas, shows a significant increase between 4 wk and 8 and 10 wk (black bar and asterisk). For CD25 null corneas, axon density increases between 4 and 6wk (grey bar and asterisk); at 8 and 10/11 wk, axon densities for CD25 null corneas are not significantly different from those seen at 4 or 6 wk by ANOVA. By *t* test, CD25 null axon density is significantly reduced as compared to wt at 4, 8, and 10/11 wk (maroon bar and asterisk). Combining axon densities for all ages confirms that CD25 null corneas have significantly reduced axon density. (**B**) Corneal epithelial cell proliferation was determined by quantifying the numbers of ki67+ cells per field. The numbers of corneas used for each assessment are shown beneath each bar. Proliferation data for male and female corneas were combined. Using ANOVA to compare proliferation for all ages (4, 6, 8, 10/11) for wt corneas shows no significant difference between 4 and 6 wk; however, between 6 and 8 wk, there is a reduction in cell proliferation (black bar and asterisk). Between 8 and 10/11 wk, cell proliferation does not change. Using ANOVA to compare proliferation for all ages (4, 6, 8, 10/11) for CD25 null corneas shows no significant difference at any time point. By *t* test, CD25 null proliferation is significantly elevated as compared to wt at 4 and 10/11 wk (maroon bar and asterisk). Combining cell proliferation data for all ages confirms that CD25 null corneas have significantly elevated cell proliferation compared to wt. (**C**) To determine how corneal epithelial cell proliferation varies with axon density, we calculated Spearman correlation coefficients for the paired values for axon density and cell proliferation from individual wt and CD25 null male and female corneas. Data for both sexes were combined. Data shows that in wt corneas, there is a significant negative correlation between cell proliferation and axon density at 6 and 10/11 wk and in CD25 null corneas there is a significant negative correlation between cell proliferation and axon density at 4 and 8 wk. Combining data for all ages confirms the negative correlation between cell proliferation and axon density for both genotypes.

**Figure 4 ijms-19-03821-f004:**
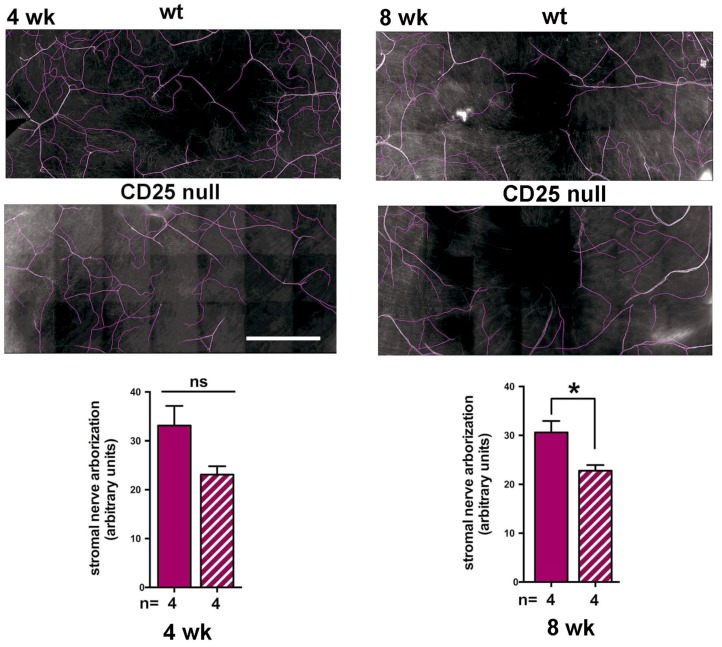
Stromal nerve arborization is reduced in CD25 null corneas. Whole flat mount corneas stained with βIII tubulin were imaged below the level of the ICNs within the stroma; 4 corneas from 4 and 8 wk female mice were imaged for each variable assessed. Stromal nerve arborization was traced using NeuronJ. Shown are representative images obtained at 4 and 8 wk of age in wt and CD25 null corneas. In wt corneas at 4 wk, stromal nerve arborization is more variable between individual corneas than at 8 wk. Although stromal nerves are less abundant at 4 wk in CD25 null corneas (hatched bar), the difference is not significant. By 8 wk, the variability in the wt corneas is reduced and the difference in the null corneas is significant. Bar = 500 µm.

**Figure 5 ijms-19-03821-f005:**
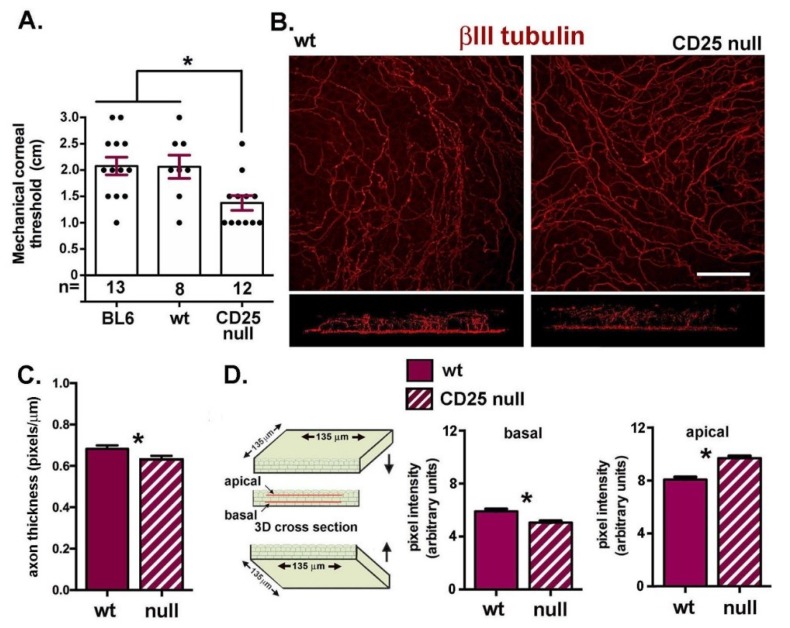
Corneal sensitivity is reduced and differences are seen in ICN thickness and intraepithelial nerve terminal (INT) apical extension in CD25 null corneas. (**A**) Corneal sensitivity was assessed using mechanical stress threshold. The numbers of corneas assessed for each variable are indicated beneath each bar. CD25 null corneas had significantly lower stress threshold compared to C57BL6 or CD25 wt mice. (**B**) To assess INT apical extension, 40x en face confocal image stacks showing ICNs were generated and rotated to create a cross sectional view. Representative images are shown; a minimum of four corneas were assessed for each variable assessed. (**C**) Axon thickness was determined on *n* = 4 corneas for 4 and 8 wk female wt and CD25 null mice. CD25 null corneas had significantly thinner axons. (**D**) Cross-section projection views through 135 µm of corneal epithelium were generated and pixel intensities determined across two lines; one above the sub-basal nerve density and a second beneath the apical-most cells. Data show that CD25 null corneas have significantly lower pixel intensities basally, but higher pixel intensities apically. While fewer INTs branch from the sub-basal axons in CD25 null corneas, those that do extend apically and can branch more within apical cells as compared to wt corneas. Bar in B = 35 µm.

**Figure 6 ijms-19-03821-f006:**
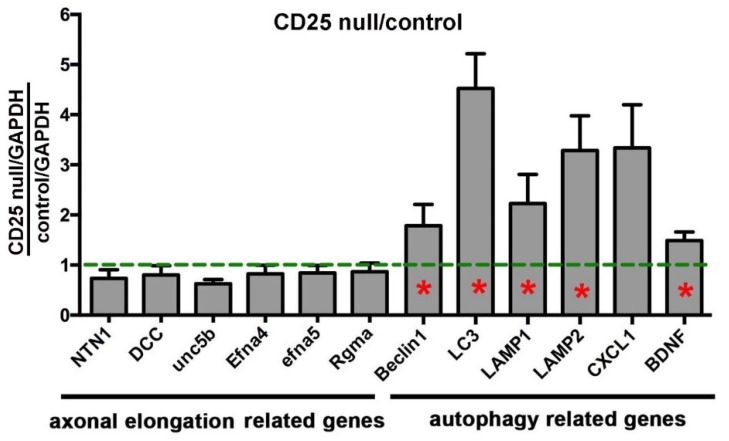
Expression of mRNAs for proteins that mediate autophagy and phagocytosis are increased in corneal epithelium of CD25 null mice. qPCR studies were performed on mRNA isolated from the corneal epithelium of female 8 wk old wt and CD25 null mice. Data for CD25 null mice after normalization against mRNA expression relative to wt mice are shown. Red asterisks within bars indicate significant differences in RNA expression relative to RNA isolated from wt mice. There are no significant differences seen in expression of mRNAs for proteins that enhance or reduce axon targeting including netrin1, its receptors DCC, Unc5b, and Efna5, as well as Repulsive guidance molecule A (Rgma). By contrast, mRNAs for proteins activated during phagocytosis and autophagy are increased (Beclin1, LC3, LAMP1, LAMP2, and Brain derived neurotrophic factor (BDNF)).

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
