# Peer review of "Reduced Corneal Innervation in the CD25 Null Model of Sjögren Syndrome"

_ijms, 2018, doi:10.3390/ijms19123821_

Reviewer 1 Report

Although the observations of decreased anatomic corneal innervation accompanied with reduced corneal sensitivity, increased corneal epithelial cell proliferation, and increased expression of genes regulating phagocytosis and autophagy in CD25 null Sjögren syndrome model are very interesting, a number of points need clarifying and certain statements require further justification. These are given below.

1.    Figure 2 seems apparently unnecessary because the description (lines 111-115) is sufficient for readers to understand no significant differences between sexes in wild and CD25 null axon density.

2.    In Figure 4, the authors analyzed using NeuronJ software. The “NeuronJ” plugin software is free for download and usage in the condition citing the paper of Meijering, E. et al(Cytometry Part A58,167-176, 2004: see https://imagescience.org/meijering/software/neuronj/). However, the authors did not describe that they analyzed their data using NeuroJ software in Method section. In addition, they did not cite the paper of Meijering, E. et al.

3.    The papers concerning Sjögren syndrome model in CD25 knockout mice (De Paiva, C.S. et al.Rheumatology49,246-258, 2010; Pelegrino, F.S.A. et al. Arthritis Res. Ther.14,R234, 2012) should be included.

4.    CD25 mRNA was expressed in C57 corneal epithelium, but not in Balb/c corneal epithelium. The authors should describe what about in human corneal epithelium and discuss including human cornea.

Typographical errors listed below should be corrected:

Line 30: Sjögren Syndrome -> Sjögren syndrome

Line 39: [2-6] ; a recent study -> [2-6]; a recent study

Line 46: permeability , increased infiltration -> permeability, increased infiltration

Line 160: Neuron J -> NeuronJ

Line 214: QPCR -> qPCR

Line 230: QPCR -> qPCR

Line 349: QPCR -> qPCR

Line 350: QPCR -> qPCR

Line 352: QPCR -> qPCR

Line 356: QPCR -> qPCR

Line 362: QPCR -> qPCR

Line 363:6. Statistical analyses-> 6. Statistical analyses

Line 366:A pvalue -> A pvalue

Line 385: Autoimmunity reviews-> Autoimmunity Reviews

Line 488: Frontiers in immunology -> Frontiers in Immunology

Lines 394-395: The Journal of allergy and clinical immunology -> The Journal of Allergy and Clinical Immunology

Line 397: Clinical immunology -> Clinical Immunology

Line 399: Clinical and experimental immunology -> Clinical and Experimental Immunology

Lines 401-402: Eye & contact lens -> Eye & Contact Lens

Line 404: Trends in immunology -> Trends in Immunology

Line 406: Journal of immunology -> Journal of Immunology

Line 409: Journal of immunology -> Journal of Immunology

Line 412: Mucosal immunology ->Mucosal Immunology

Lines 414-415: official journal of the Controlled Release Society -> Official Journal of the Controlled Release Society

Line 417: The American journal of pathology -> The American Journal of Pathology

Line 423: Arthritis research & therapy -> Arthritis Research & Therapy

Line 427: Acta ophthalmologica -> Acta Ophthalmologica

Lines 429-430: Investigative ophthalmology & visual science -> Investigative Ophthalmology & Visual Science

Line 432: Acta ophthalmologica -> Acta Ophthalmologica

Line 435: The Journal of comparative neurology -> The Journal of Comparative Neurology

Line 439: Investigative ophthalmology & visual science -> Investigative Ophthalmology & Visual Science

Line 443: Graefe's archive for clinical and experimental ophthalmology -> Graefe's Archive for Clinical and Experimental Ophthalmology

Line 447: Journal of allergy and clinical immunology -> Journal of Allergy and Clinical Immunology

Line 449: Investigative ophthalmology & visual science -> Investigative Ophthalmology & Visual Science

Line 453: Experimental eye research -> Experimental Eye Research

Line 456: Experimental eye research -> Experimental Eye Research

Line 458: Experimental eye research -> Experimental Eye Research

Lines 460-461: Investigative ophthalmology & visual science -> Investigative Ophthalmology & Visual Science

Line 463: Journal of inflammation -> Journal of Inflammation

Line 469: Advances in wound care -> Advances in Wound Care

Line 471: PLoS biology -> PLoS Biology

Line 474: PloS one -> PLoS One

Line 476: Trends in immunology -> Trends in Immunology

Lines 479-480: Current opinion in immunology -> Current Opinion in Immunology

Lines 481-482: The FEBS journal -> The FEBS Journal

Line 487: Scientific reports ->Scientific Reports

Line 490: Laboratory investigation -> Laboratory Investigation

Line 493: Investigative ophthalmology & visual science -> Investigative Ophthalmology & Visual Science

Author Response

We want to thank the reviewer for the time taken to review our manuscript and to suggest ways to improve it.  We have made the changes requested as described below.

1.    Figure 2 seems apparently unnecessary because the description (lines 111-115) is sufficient for readers to understand no significant differences between sexes in wild and CD25 null axon density.  Our major funding agency (NIH) has asked that we explicitly address sex as an independent variable in the studies they support.  We could remove this figure but many readers prefer visual presentation of data.  Including Figure 2 makes it easy for people to see that we take seriously the NIH mandate to investigate whether sex differences contributed to the differences reported in these studies. They did not.

2.    In Figure 4, the authors analyzed using NeuronJ software. The “NeuronJ” plugin software is free for download and usage in the condition citing the paper of Meijering, E. et al(Cytometry Part A58,167-176, 2004: see https://imagescience.org/meijering/software/neuronj/). However, the authors did not describe that they analyzed their data using NeuroJ software in Method section. In addition, they did not cite the paper of Meijering, E. et al.  We regret not having cited that reference and link to NeuronJ in the original submission.  We have added the link to thesite where NeuronJ can be downloaded in the methods section and have added the citation by Meijering and colleagues to the reference list.

3.    The papers concerning Sjögren syndrome model in CD25 knockout mice (De Paiva, C.S. et al.Rheumatology49,246-258, 2010; Pelegrino, F.S.A. et al. Arthritis Res. Ther.14,R234, 2012) should be included.  We had already cited the De Paiva, et al., Rheumatology paper from 2010; we have now added the paper by Pelegrino and colleagues from 2012.  We revised the text to read as follows:  “While CD25 has been reported to be expressed on conjunctival and corneal epithelia in the C57BL/6 mouse strain [29, 30] and on human ocular surface epithelia [30], RNAseq studies show that CD25/IL2RA mRNA is undetectable in the Balb/c unwounded corneal epithelium and stroma [31].”    

4.    CD25 mRNA was expressed in C57 corneal epithelium, but not in Balb/c corneal epithelium. The authors should describe what about in human corneal epithelium and discuss including human cornea.  We have cited the paper by De Paiva and colleagues (De Paiva CS, Yoon KC, Pangelinan SB, et al.: Cleavage of functional IL-2 receptor alpha chain (CD25) from murine corneal and conjunctival epithelia by MMP-9. Journal of inflammation 6: 31, 2009) that shows CD25 in human corneal and conjunctiva epithelial cells.  

The corrections requested have been made.

Reviewer 2 Report

Authors present interesting data obtained in a mice model of Sjogren's syndrome. Corneas from both CD25 null and control animals were analyzed with respect to mechanical sensitivity , immunofluorescence (βIII tubulin and ki67) for detection of intraepithelial nerve terminals, confocal microscopy, QPCR for analysis of the expression of genes regulating phagocytosis and autophagy. Results are elegantly presented and consistently demonstrate that decreased corneal innervation in the null CD25 mice model is in relation with reduced corneal sensitivity, increased corneal epithelial cell proliferation and expression of genes regulating phagocytic processes. The issue paper is of interest for the scientific community, the science and the design of the project as well as the quality of the manuscript are of really good level.  No specific revision is requested and the present reviewer recommends the publication of the manuscript as it is. 

Author Response

We appreciate the comments supportive of the data presented in this study.   No revisions were requested. 

Reviewer 3 Report

The authors study in an animal model of SS the changes in corneal sensory nerve morphology and dysfunction.

It is a well-designed paper but two suggestions have been made:

In fig 4 there are two graphics between the photographs that need to improve the information

In the discussion it has to be described the reason to take just 4 corneas in some of the experiments, where do they come from male or female specimens? and the differences between the proportion of sex in the groups

Author Response

In fig 4 there are two graphics between the photographs that need to improve the information. We had included images with and without the stromal axons traced using NeuronJ.  We have removed the replicate images and only show the images with the magenta tracings of the stromal axons.

In the discussion it has to be described the reason to take just 4 corneas in some of the experiments, where do they come from male or female specimens? and the differences between the proportion of sex in the groups.  The only experiments where we used 4 corneas were the stromal nerve arborization experiments presented in Figure 4.  Because stromal nerve patterns are less variable between individual corneas, we needed fewer replicates to achieve significance.  The rest of the studies presented in other figures use many more corneas.  We use female 4 and 8 wk corneas for these experiments. To clarify, we added the following statement to the Methods section:

“We used female mice at 4 and 8 wk of age; stromal nerve arborization data are less variable between individual corneas than sensory nerve density data which allowed us to use fewer corneas for these assessments.”